# The Role of Surgery in Primary Chest Wall Tumors: Over 20 Years’ Experience in Resection and Reconstruction

**DOI:** 10.3390/cancers15072153

**Published:** 2023-04-05

**Authors:** Giorgio Lo Iacono, Antonio Mazzella, Shehab Mohamed, Francesco Petrella, Giulia Sedda, Monica Casiraghi, Lara Girelli, Luca Bertolaccini, Lorenzo Spaggiari

**Affiliations:** 1Department of Thoracic Surgery, IEO, European Institute of Oncology IRCCS, 20141 Milan, Italylorenzo.spaggiari@ieo.it (L.S.); 2Department of Oncology and Hemato-Oncology, University of Milan, 20122 Milan, Italy

**Keywords:** chest wall, sternum, ribs, chondrosarcoma, sarcoma, reconstruction

## Abstract

**Simple Summary:**

Primary chest wall tumors comprise a group of rare tumors for which the best treatment method has yet to be determined. As surgeons, we wanted to explore our role in approaching these pathologies to analyze the role of surgery amidst other oncological considerations. It is clear that radical oncological surgery is an essential element of a multidisciplinary approach.

**Abstract:**

Background: Primary chest wall tumors comprise a heterogeneous group of neoplasms arising from soft tissues and bones. While surgical excision is the standard of care for benign tumors, the management of malignant tumors requires multimodal treatment. We conducted a predictive analysis of outcome, recurrence-free and overall survival. Methods: We retrospectively reviewed the clinical and pathological records of all patients treated in our center between 1998 and 2020. Results: 53 patients (15–85 years) were treated in our department. The average tumor diameter was 65 ± 35 mm (10–160 mm). Negative margins were obtained in 48 patients (90.6%), whereas in the remaining 5, R1 resection was accomplished. Median overall survival was 63,03 months (1–282 months). Overall survival was 90% at 1 year, 78% at 2 years, and 61% at 5 years. Our analysis identified tumor diameter, postoperative complications, and high grade of malignancy as factors that can influence prognosis. Conclusions: The treatment of primary chest wall tumors remains a very challenging process. Different histological types preclude definition of an unequivocal approach. Complete resection with healthy margins remains a definitive cornerstone in the treatment of these cancers as part of a more comprehensive approach.

## 1. Introduction

It is believed that Osias Aimar was the first to perform, in 1778 [1], the resection of a chest wall tumor. Primary chest wall tumors comprise a heterogeneous group of neoplasms arising from soft tissues and bones of the chest. They are rare tumors, representing around 5% of all thoracic neoplasms [1,2,3]. They are classified based on histology type and clinical behavior. Benign lesions arising from the thoracic district are more often represented by Osteochondromas (50%), Chondromas, Fibrous dysplasia, and Desmoid tumors [1].

Sarcomas dominate malignant pathology. Chondrosarcomas are the most common primary chest wall tumors, followed by Ewing sarcoma and Askin’s tumors [4]. Patients typically refer to the appearance of a mass in the chest or local pain. The non-specificity of initial symptoms can be a barrier to early diagnosis. Usually these symptoms are often linked to the local invasion of adjacent structures. Due to the rarity of these tumors, most studies have collected a limited number of cases, even over long time intervals. Over the years and based on the histotypes, various combinations have been used between surgery, neoadjuvant and adjuvant chemotherapy, and radiotherapy, with variable results [5,6,7,8]. Over the years, surgery has found a fundamental space in the treatment of these cancers. While surgical excision is the standard of care for benign tumors, the management of malignant tumors requires multimodal treatment. Due to the relative chemo- and radio-resistance of some of these tumors, the key to success has been seen to be a resection with large healthy margins on both the bony and soft tissue parts, especially in the more aggressive sarcomas. We aimed to analyze which factors affect early and long-term outcomes as well as which factors have a crucial role in relapse; above all, we sought to determine the best approach for these neoplasms. We analyzed our 22 years of experience in this field for these reasons.

## 2. Materials and Methods

We retrospectively evaluated the clinical and pathological findings of all patients affected by primary chest wall malignancy treated in our center between 1998 and 2020. The research was conducted according to the recommendations of the Declaration of Helsinki. All patients authorized, through written informed consent, the use of their data for scientific purposes. The Ethics Committee and the Internal Review Board, informed of the database extraction, did not require approval because of the study’s retrospective nature. This manuscript was written according to the Strengthening the Reporting of Cohort Studies in Surgery (STROCSS) Statement [9]. The STROCSS checklist is available as Appendix A.

### 2.1. Preoperative Assessments

Routine preoperative investigations include chest X-ray, whole-body CT scan, or magnetic resonance imaging if needed. A positron emission tomography (PET) scan has been routinely performed since 2003. The functional assessment includes spirometry and preoperative echocardiography with the assessment of left ventricular ejection fraction. If possible, a Tru-Cut biopsy/Core biopsy or surgical incisional biopsy is performed during the preoperative assessment. Indications for neoadjuvant or adjuvant treatment in our institution are discussed individually during a weekly Multidisciplinary Tumor Board. Medical Oncologists decide on chemotherapy regimens according to current guidelines. In the case of positive margins (R1), patients receive radiotherapy as adjuvant treatment according to the Multidisciplinary Tumor Board, with a plan decided by Radiation Oncologists according to current guidelines.

### 2.2. Collected Data

Preoperative and operative records include: induction chemotherapy or radiotherapy, type of operation, histologic type, grading of malignancy, the diameter of tumor, number of resected ribs, type of reconstruction, oncological radicality, type of adjuvant therapy, as well as short- and long-term outcomes. All the data were retrospectively collected. Postoperative outcomes include ICU stay, discharge after surgery, and postoperative complications according to Clavien-Dindo classification [10], as well as the date and site of relapse, if present. Patients with metastatic tumors of the chest wall or direct invasion from other sites were excluded. We reviewed pathological findings, surgical approach, preoperative and postoperative treatment, and early and long-term results. The follow-up duration was defined from the date of the operation.

### 2.3. Statistical Analysis

For data processing and analysis, IBM SPSS Statistics 21.0 software (IBM, Armonk, NY, USA) and RStudio (R version 4.2.1, Funny-Looking Kid) (Team R. RStudio: Integrated development environment for R. Boston, MA, USA: RStudio, Inc.; (2021)) were used for statistical analysis. Results are expressed as the percentage for qualitative parameters and mean and standard deviation (SD) or median (interquartile range) for normally and non-normally distributed quantitative variables, respectively. Follow-up information was obtained by telephone interviews with patients or relatives and referring physicians. Patient overall survival was assessed by the Kaplan–Meier method, using the date of treatment of the primary tumor as time 0. The end of the follow-up was 31 December 2021. The log-rank test compared survival rates, and Bonferroni correction was applied for multiple comparisons. Variables significantly associated with survival at univariable analysis were subjected to Cox multivariable analysis to assess their independent character. The cut-off of 65 mm was chosen based on the mean tumor size; the cut-off of 2 ribs was similarly chosen based on the mean value of resected ribs. A *p*-value of less than 0.05 was considered significant.

## 3. Results

A total of 53 patients (25 men and 28 women) were treated in our department for primary chest wall malignancy between 1998 and 2020 (Figure 1). The median age was 46 years (range, 15 to 85 years).

In 19 patients, a FNAB was performed by CT scan. In 15 patients, a preoperative incisional biopsy was performed for the purposes of diagnosis. In the remaining patients, an intervention was carried out without a preoperative diagnosis.

The final pathologic exam revealed: 24 chondrosarcomas (45.3%), 4 osteosarcomas (7.6%), 10 Ewing/Askin sarcomas (18.9%), 3 angiosarcomas (5.7%), 5 synovial sarcomas (9.5%), 2 rhabdomyosarcomas (3.7%), 1 leiomyosarcoma (1.9%), 2 liposarcomas (3.7%), and 2 plasmacytomas (3.7%).

Neo-adjuvant treatment consisted of chemotherapy (15 patients) or a combination of radio- and chemotherapy (1 patient), followed by chest wall resection. In the remaining 37 patients, surgery was the initial therapeutic approach.

Chest wall resections included resected ribs in 43 patients. Partial (12 patients) or total (3 patients) sternal resection was performed in 15 patients out of 53. Surgical resection was extended to the lung in 16 patients (12 wedge resections, 3 lobectomies, 1 pneumonectomy), to the clavicle in one patient, and to the scapula in two patients (one partial and one total resection). The average diameter tumor was 65 ± 35 mm (range: 10–160 mm); the median number of resected ribs was 2 (range: 0–6). Adjuvant chemotherapy was performed in 6 patients, radiotherapy in 10 patients, and a combination of both in 12 patients. Negative margins (R0) were obtained in 48 patients (90.6%), whereas the remaining 5 R1 resections were accomplished. We did not have any R2 resections. To cover the prosthesis, in 20 patients, we used a rotational muscle flap (15 latissimus dorsi, 5 pectoralis major).

The median follow-up time was 81.3 months. The median overall survival was 63.03 months (1 to 282 months). Overall survival was 90% at 1 year, 78% at 2 years, and 61% at 5 years (Figure 2).

Univariable analysis of factors possibly affecting survival showed that the diameter of resected tumor > 65 mm (*p* = 0.04), number of resected ribs > 2 (*p* = 0.016), ICU stay (*p* = 0.038), neo-adjuvant treatments (*p* = 0.036), and adjuvant treatments (*p* = 0.014) were negative predictors, as well as the presence of postoperative complications (*p* = 0.023). The results of the univariable analysis of the other variables are reported in Table 1.

Univariable analysis of factors possibly affecting relapse showed that the diameter of resected tumor > 65 mm (*p* = 0.045), number of resected ribs > 2 (*p* = 0.015), high grade of malignancy (*p* = 0.023), and adjuvant treatments (*p* = 0.026) were negative predictors. The results of the univariable analysis of the other variables are reported in Table 2.

The multivariable analysis for survival confirmed that number of resected ribs (HR = 1.58; 95% confidence interval (CI): 1.31–2.48, *p* = 0.045) and adjuvant treatments (HR = 1.64; 95% CI: 1.37–2.18, *p* = 0.047) were independent prognostic factors (Table 3, Figure 2).

The multivariable analysis for the relapse showed that the only independent prognostic factors were the number of resected ribs >2 (HR = 1.58, 95% CI: 1.23–2.13, *p* = 0.048) and adjuvant treatments (HR = 2.32; 95% CI: 1.23–2.67, *p* = 0.046) (Table 4, Figure 3).

## 4. Discussion

Primary chest wall tumors have historically represented a great challenge for surgeons.

Small series, huge masses, and high levels of morbidity and mortality, until recently, have resulted in limited success in treatment. Surgery has been recognized as a fundamental step in the multimodal approach to this kind of tumor due to several histotypes’ relative chemo- and radio-resistance. The major part of primary chest wall tumors is superficial, and even if highly malignant, there is favorable prognosis compared to the same histotypes localized in the retroperitoneal space. Like most studies on this pathology, the limitations of this study are its retrospective nature, the lack of a control group, and the significant heterogeneity of the lesions. However, we believe that the long observation time interval and the sample size are worth analyzing to obtain information to guide the treatment of these tumors. Usually, patients refer to a growing mass in the chest; less commonly, the diagnosis is linked to an incidental finding studying an unrelated condition. Soft tissue masses are often painless, while lesions rising from the bones due to periosteal invasion are typically painful. Symptoms are often related to the local invasion of adjacent structures. Systemic manifestations, such as fever, weight loss, and malaise, are frequently related to more advanced diseases.

There are no specific features to distinguish a benign from a malignant lesion with only clinical examination because many features are present in both types. In addition to the clinical evaluation, the preoperative staging provided for all patients was a study with a Total Body CT scan, a PET FDG, and a complete cardiorespiratory study. In some cases, an MRI was used to study some details in certain districts, but we did not use it routinely. The same methods were also used for restaging in the case of induction therapy. The combination of CT and PET scan has been proven to increase the reliability of staging and restaging; it is also used in follow-up [11]. The appropriate approach for the treatment of primary chest wall tumors is to obtain an accurate histological diagnosis. Ideally, a preoperative diagnosis should be obtained in all patients with a chest wall mass. In cases of small lesions, an excisional biopsy can represent a traditional approach. Surgical resection must ensure healthy margins to reduce the risk of local recurrence. For benign tumors, “negative margins” may be sufficient, but for aggressive tumors, up to 4 cm of wide margins may be required [3]. In the absence of histology for direct resection of the lesions, margins of at least 2 cm should be respected.

Incisional biopsy is especially useful for large masses to define the operative strategy and FNAB, the latter preferably with a large needle to obtain a tissue sample suitable for the correct histopathological definition. During definitive surgery, it is essential to remove the site of the previous biopsy to avoid the risk of seeding, especially in high-grade tumors. For non-palpable lesions, labeling with radioactive elements or dyes, which can be easily found during surgical excision, should be considered to minimize the risk of error. 

There is no evidence on the role of lymphadenectomy, but it should be performed in the case of clear evidence of node involvement [11,12].

The possibility of obtaining a radical resection often clashes with the size and extent of the disease, as well as with the site of presentation. It is more complex to achieve oncological radicality in some areas of the thorax, such as proximity to the spine or invasion of the lung or mediastinum. However, the extension of the disease must not be sacrificed in any way with the aim of oncological radicality for fear of too extensive a resection. Today, the availability of new prosthetic materials and the wide availability of myocutaneous flaps allow reconstruction even in huge masses. Collaboration with teams of orthopedists and plastic surgeons can significantly improve surgery outcomes.

Reconstruction has always been performed by choosing the technique based on the site and the extent of the resection. In two cases, we used Goretex mesh; in 13 cases, the choice was a Vicryl mesh; in two other cases, we did not place any prosthesis due to the site of resection. In the rest of our series, a double Marlex mesh with methyl methacrylate was used for the reconstruction (Figure 4 and Figure 5). 

Although not fashionable in recent years, due to the use of new devices and materials [13,14,15], this technique has allowed delivered excellent outcomes, including customization in accordance with wall defects and good functional, rigidity, and aesthetic results. Finally, we found a meager rate of prosthetic infection, to the point that only one prosthesis was removed for infection out of a total of 32. We also highlighted these reliability and efficiency data in another series published by our group in about 166 cases of patients with all tumor types, primary and secondary, who underwent chest wall resection and rigid prosthesis reconstruction over 18 years, where only 8 patients required prosthesis removal for infection [16,17].

While small posterior and apical chest wall defects generally do not require reconstruction for the natural protection of the rest of the chest wall, lateral and anterior resection always need to be reconstructed due to the risk of lung herniation [18]. The impaction of the scapula in posterolateral resection should be considered. In these cases, a synthetic mesh must be placed. In 20 patients, it was necessary to proceed with reconstruction with a rotational muscle flap to cover the prosthesis. The most used muscle was the latissimus dorsi in 15 cases, while in the remaining 5 cases, we created a pectoralis major muscle flap. Our plastic surgeons performed all reconstructions.

We did not encounter perioperative mortality. Minor complications were reported in 26% of cases (we report only atrial fibrillation and postoperative anemia). Only 4 patients out of 53 (7.5%) experienced major complications: 1 hemothorax requiring reoperation for hemostasis, 1 flap necrosis, 1 flap infection, and 1 pulmonary embolism. A series published in 2017 showed lower blood loss during surgery as a predictor of better DFS [19]. Although we did not collect this data for evaluation, as we were not concerned with significant losses during surgery, it might represent data to consider for future work.

We do not report respiratory complications. We believe that early activation of physical and respiratory physiotherapy activities, starting from the first postoperative day, is fundamental for the prevention of respiratory complications.

Following the increasingly widespread development of minimally invasive techniques in thoracic surgery, several groups have experimented with the application of these techniques, both with the long-term VATS technique [20] and the robot, especially for combined resections on the lung parenchyma. Despite our extensive experience with these techniques, we have never found a clear advantage in the field, except in a single case of a Pancoast tumor [21]. Furthermore, the literature in this regard does not seem to highlight a clear advantage in terms of complications or morbidity [22], especially in frail patients.

In our study, the mean diameter of the removed tumors was 65 mm, and the analysis showed that the increase in size and, therefore, the relative number of removed ribs are two of the parameters that influence survival and relapse, and they refer to pathological aggressiveness.

Concerning the role of the adjuvant therapy (RT or RT + CT) in our analysis, its prognostic value should be considered as the result of the aggressiveness of the tumor; indeed, in the case of post-surgical R1 (5 patients), more aggressive grading (>1), or more significant dimensions, patients underwent multimodal treatment, according to the Institutional policy of our Multidisciplinary Tumor Board. Thus, considering the notorious radio- and chemo-resistance of sarcomas, the negative prognostic value of adjuvant therapy in our cohort should be considered an indirect sign of the more aggressive behavior of the sarcomas.

In our analysis, only five patients presented R1 post-surgical resection, which did not impact survival or relapse. In addition, we performed in all five patients adjuvant radiotherapy (three cases) or a combination of chemo- and radiotherapy (two cases) to minimize the effects on survival or relapse.

While neoadjuvant therapy is rarely practiced because its use does not seem to have a significant impact on resection or survival [23] except in selected cases, adjuvant therapy is used more frequently and can be reasonably standardized for some histotypes (Ewing, Askin, Osteosarcoma), though it is less frequently used in soft tissue tumors, regardless of grading [2].

A meta-analysis confirms a marginal effect of chemotherapy in localized resectable soft-tissue sarcoma concerning overall survival, recurrence-free survival, and local and distant recurrence [6].

Radiotherapy, on the other hand, is indicated in cases of incomplete R1 or R2 resection or where the extension of the resection cannot be further practiced.

### Strength and Limitations

Our study has several limitations. It includes a retrospective series of patients collected prospectively. Additionally, a relatively small number of patients is involved. Our findings should therefore be regarded with caution. The primary advantage is the uniform series of procedures conducted over time with unaltered indications for surgery and an oncologic care plan. In addition, the supplied statistics derive from the 20-year experience of a high-volume referral center, from which significant conclusions can be inferred.

## 5. Conclusions

Despite the progress in the treatment of primary chest wall tumors, even with a multimodal approach, it remains a very challenging pathology. Different histological types prevent the definition of an unequivocal multimodal strategy. It is widely accepted that surgery plays a vital role in the treatment of these cancers, both as an exclusive treatment and as part of a more comprehensive plan. Indeed, complete resection with healthy margins remains a definitive cornerstone.

Our series suggests that dimensions of the lesion and subsequent extent of surgical resection play an essential role in survival or relapse. Even if industrial progress have given us new materials with which to reconstruct the defect after resection, methyl methacrylate remains a haven even in significant reconstruction. Given the particularity of the surgery and the rarity of the presentation, patients must be referred and treated in high-volume centers with more experience in the field and in the presence of a multidisciplinary team, which can provide the patient with all the necessary tools for care in all its aspects.

## Figures and Tables

**Figure 1 cancers-15-02153-f001:**
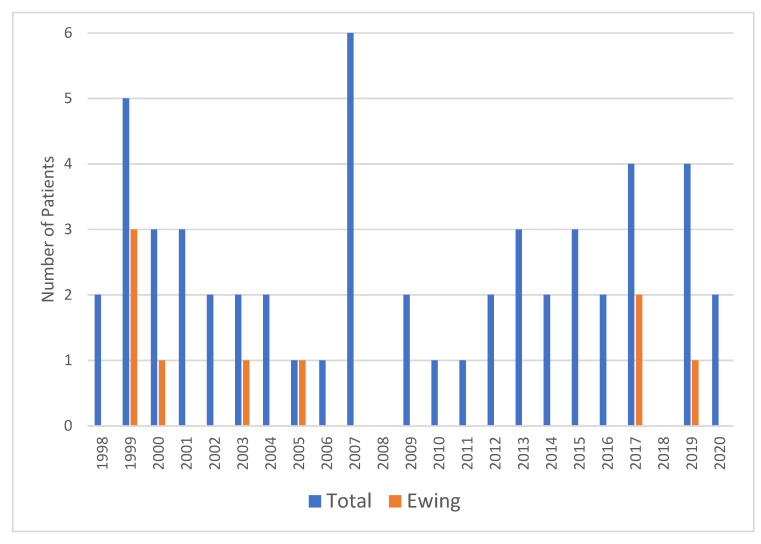
Grouped bar plot with annual cases and Ewing’s sarcoma cases over time.

**Figure 2 cancers-15-02153-f002:**
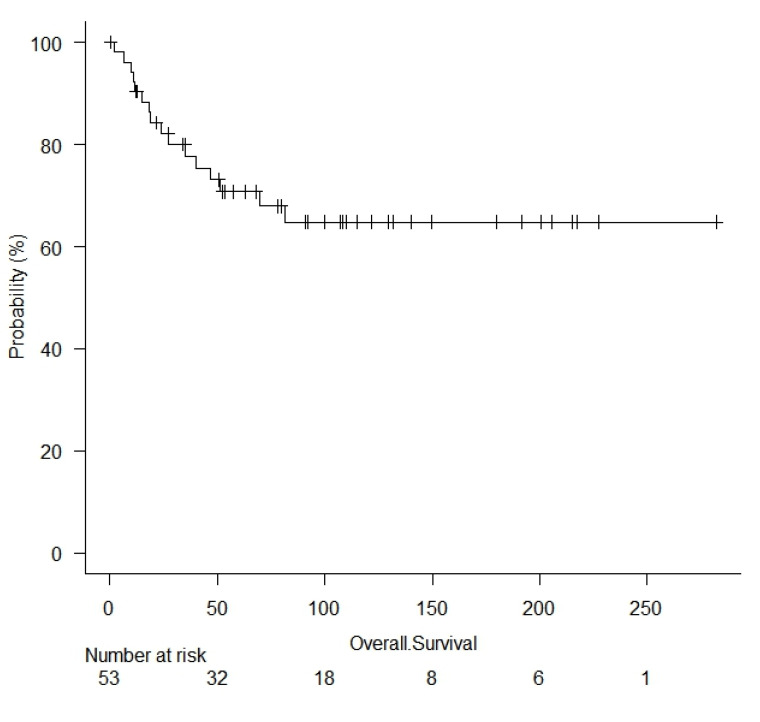
Kaplan–Meyer curves for overall survival; survival for adjuvant treatments (log rank = 0.00057), the number of resected ribs (log rank = 0.031), Ewings sarcoma (log rank = 0.019), and sternal reconstruction (log rank = 0.47). CT = chemotherapy; RT = radiotherapy.

**Figure 3 cancers-15-02153-f003:**
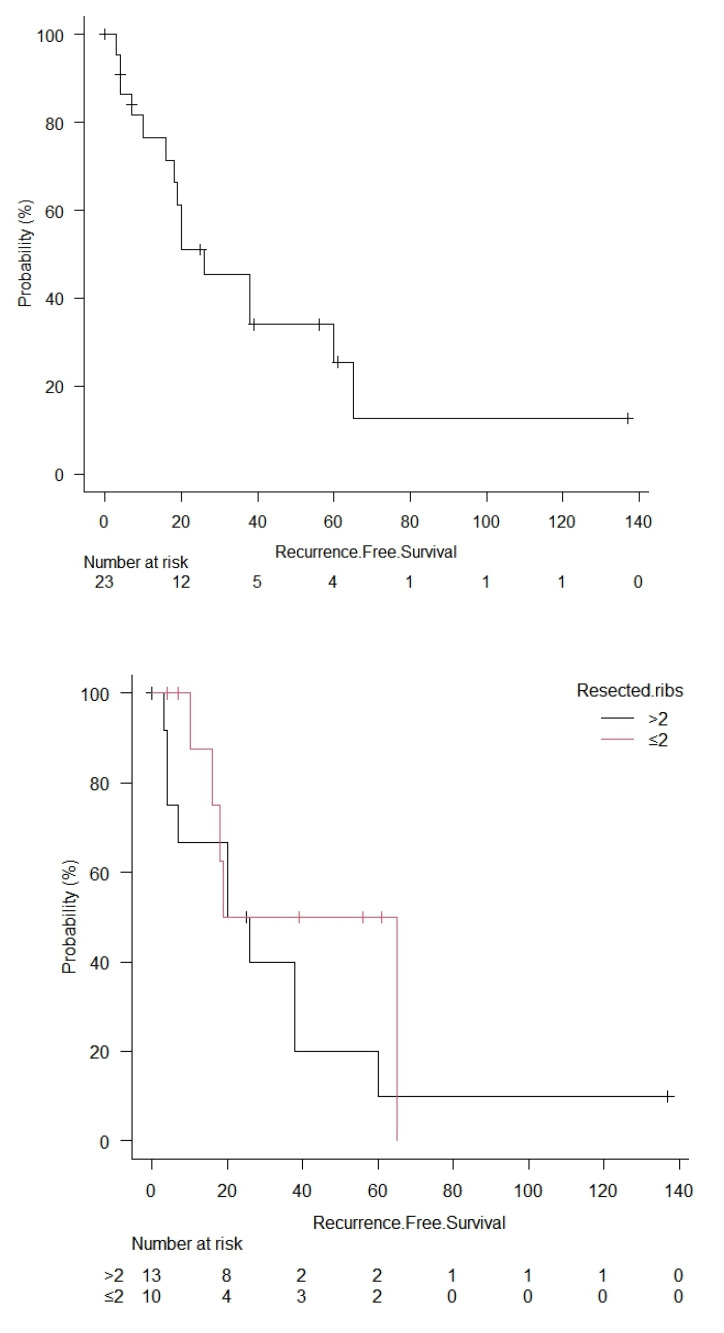
Kaplan–Meyer curve for overall recurrence-free survival; survival for the number of resected ribs (log rank = 0.35), adjuvant treatments (log rank = 0.0078), Ewings sarcoma (log rank = 0.0036) and sternal reconstruction (log rank = 0.71). CT = chemotherapy, RT = radiotherapy.

**Figure 4 cancers-15-02153-f004:**
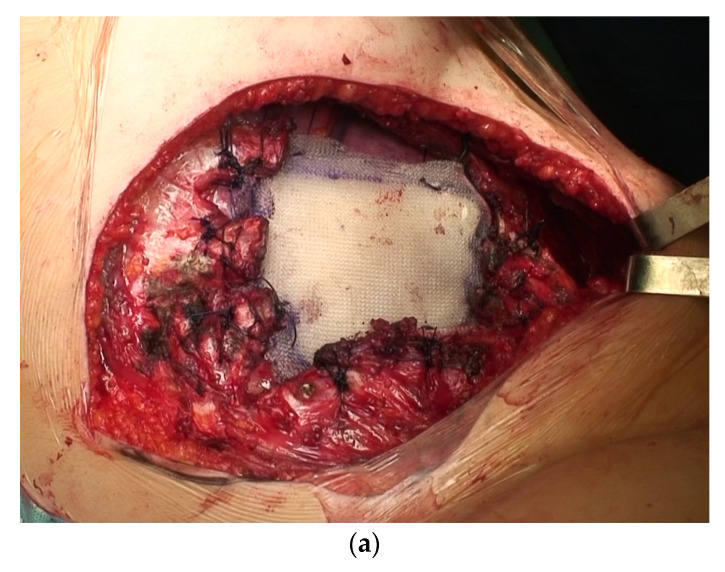
Chest wall resection and reconstruction: (**a**) reconstruction with Marlex mesh and methyl-methacrylate; (**b**) three-rib specimen with neoplasm (white).

**Figure 5 cancers-15-02153-f005:**
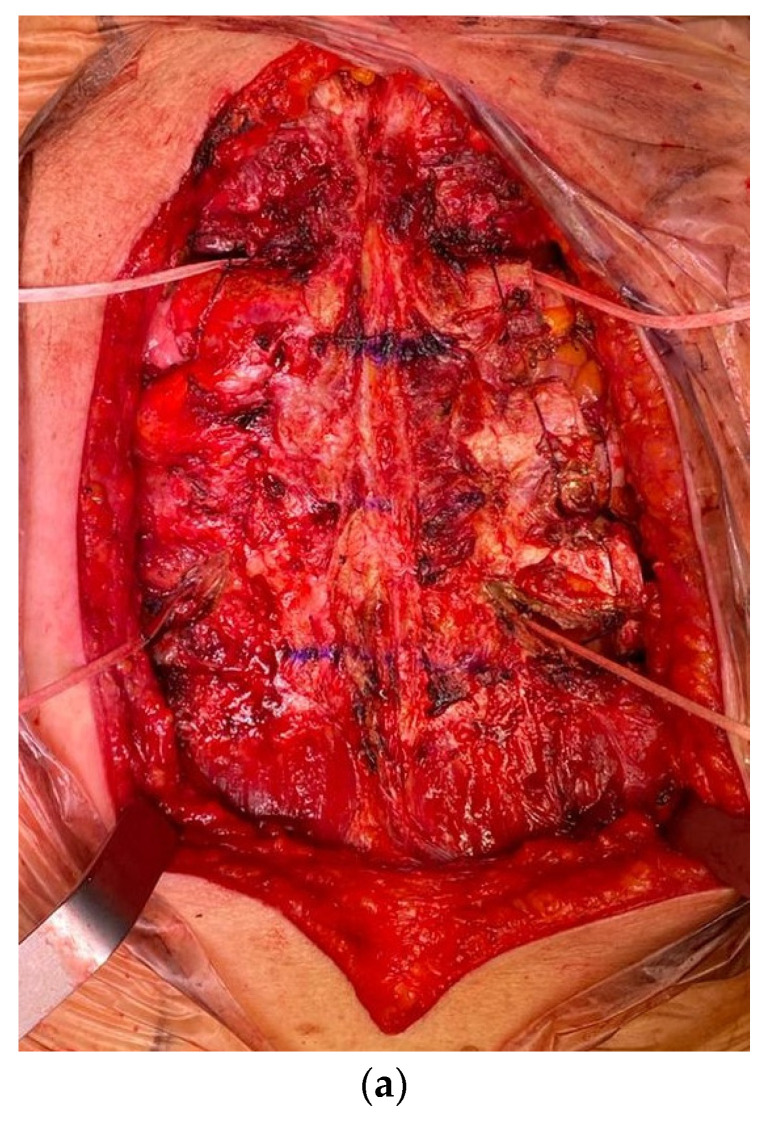
Subtotal sternectomy: (**a**) isolation of sternal plate before resection (ribs already cut); (**b**) mediastinum exposition after sternal resection; (**c**) specimen with landmarks (second, third, fourth rib, right and left. Circle: neoplasm).

**Table 1 cancers-15-02153-t001:** Univariable analysis for overall survival. Data are expressed as numbers (percentages).

Variable		Alive (No. = 37)	Deceased (No. = 16)	Log-Rank Test (*p*-Value)
Sex	M = 25 F = 28	18 (72) 19 (68)	7 (28) 9 (32)	0.49
Age	<46 = 27 ≥46 = 26	19 (70) 18 (69)	8 (30) 8 (31)	0.58
Preoperative diagnosis	Yes = 25 No = 28	10 (40) 27 (96)	15 (60) 1 (4)	0.08
Radicality	R0 = 48 R1 = 5	33 (69) 4 (80)	15 (31) 1 (20)	0.52
Grading	0–1 = 18 >1 = 35	15 (83) 22 (63)	3 (17) 13 (37)	0.11
Dimension	<65 mm = 31 ≥65 mm = 22	25 (81) 12 (55)	6 (19) 10 (45)	0.04
Resected ribs	≤2 = 33 ≥20	27 (82) 10 (50)	6 (8) 10 (50)	0.016
Sternal resection	Yes = 38 No = 15	28 (74) 9 (60)	10 (26) 6 (40)	0.26
Reconstructive prosthesis	Methyl-methacrylate = 32 Other = 21	19 (59) 18 (86)	13 (41) 3 (14)	0.058
Muscular flap reconstruction	Yes = 29 No = 24	21 (72) 16 (67)	8 (28) 6 (33)	0.44
Histology	Sarcoma = 51 Other = 2	35 (69) 2 (100)	16 (31) 0	0.23
Neoadjuvant treatment	None = 37 Chemotherapy = 15 Radiotherapy = 1	30 (81) 6 (40) 1 (100)	7 (19) 9 (60) 0	0.036
Adjuvant treatments	None = 25 Chemotherapy = 6 Radiotherapy = 10 Chemo-radiotherapy = 12	21 (84) 2 (33) 10 (100) 4 (33)	4 (16) 4 (67) 0 8 (67)	0.014
ICU stay	Yes= 38 No = 15	30 (79) 7 (47)	8 (21) 8 (53)	0.038
Hospital length of stay	<7 = 31 ≥7 = 22	24 (77) 13 (59)	7 (23) 9 (41)	0.23
30 days morbidity	Yes = 35 No = 18	28 (80) 9 (50)	7 (20) 9 (50)	0.023

**Table 2 cancers-15-02153-t002:** Univariable analysis for recurrence-free survival. Data are expressed as numbers (percentages).

Variable		Recurrence (No. = 30)	No Recurrence (No. = 23)	Log-Rank Test (*p*-Value)
Sex	M = 25 F = 28	13 (52) 17 (60)	12 (48) 11 (40)	0.36
Age	<46 = 27 ≥46 = 26	15 (55) 15 (58)	12 (45) 11 (42)	0.55
Preoperative diagnosis	Yes = 41 No = 12	21 (51) 9 (75)	20 (49) 3 (25)	0.13
Radicality	R0 = 48 R1 = 5	27 (56) 3 (60)	21 (44) 2 (40)	0.63
Grading	0–1 = 18 >1 = 35	14 (78) 16 (46)	4 (22) 19 (54)	0.023
Dimension	<65 mm = 31 ≥65 mm = 22	21 (68) 9 (41)	10 (32) 13 (59)	0.045
Resected ribs	≤2 = 33 ≥20	23 (70) 7 (35)	10 (30) 13 (65)	0.015
Sternal resection	Yes = 38 No = 15	22 (58) 8 (53)	16 (42) 7 (47)	0.48
Reconstructive prosthesis	Methyl-methacrylate = 32 Other = 21	16 (50) 14 (67)	16 (50) 7 (33)	0.19
Muscular flap reconstruction	Yes = 29 No = 24	18 (62) 12 (50)	11 (38) 12 (50)	0.27
Histology	Sarcoma = 51 Other = 2	26 (51) 1 (50)	26 (49) 1 (50)	0.23
Neoadjuvant treatment	None = 37 Chemotherapy = 15 Radiotherapy = 1	24 (65) 5 (33) 1 (100)	13 (35) 10 (67) 0	0.058
Adjuvant treatments	None = 25 Chemotherapy = 6 Radiotherapy = 10 Chemo-radiotherapy = 12	19 (76) 1 (17) 7 (70) 3 (25)	6 (24) 5 (83) 3 (30) 9 (75)	0.026
ICU stay	Yes = 38 No = 15	23 (61) 7 (47)	15 (39) 8 (53)	0.28
Hospital length of stay	<7 = 31 ≥7 = 22	18 (58) 12 (55)	13 (42) 10 (45)	0.51
30 days morbidity	Yes = 35 No = 18	20 (57) 10 (56)	15 (43) 8 (44)	0.57

**Table 3 cancers-15-02153-t003:** Multivariable analysis for overall survival (R^2^ = 0.64).

Outcomes	HR	95% CI	*p*-Value
Dimension ≥ 65 mm	1.03	0.081–1.48	0.13
Ribs > 2	1.58	1.31–2.48	0.045
Neoadjuvant treatments	1.03	0.12–1.38	0.64
Adjuvant treatments	1.64	1.37–2.18	0.047
ICU stay	1.32	0.24–1.68	0.46
Postoperative complications	1.66	0.23–1.45	0.057

**Table 4 cancers-15-02153-t004:** Multivariable analysis for recurrence-free survival (R^2^ = 0.55).

Outcomes	HR	95% CI	*p*-Value
Grading > 1	1.65	0.17–1.97	0.53
Dimension ≥ 65 mm	1.34	0.34–1.68	0.13
Ribs > 2	1.58	1.23–2.13	0.048
Adjuvant treatments	2.32	1.23–2.67	0.046

## Data Availability

Not applicable.

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
