# Peer review of "The Role of Surgery in Primary Chest Wall Tumors: Over 20 Years’ Experience in Resection and Reconstruction"

_cancers, 2023, doi:10.3390/cancers15072153_

Round 1

Reviewer 1 Report (Previous Reviewer 3)

Lo Iacono et al reported their work named "The role of surgery in primary chest wall tumors: over 20 years’ experience in resection and reconstruction " and concluded "Treatment of primary chest wall tumors remains a very challenging topic. Different histological types do not help in defining a univocal approach. Complete resection with healthy margins remains a definitive cornerstone in the treatment of these cancers as part of a more comprehensive treatment.". I have the following comments:

- Please add grouped bar plot with annual cases and Ewing's sarcoma cases over time adjacent to each other in the same figure.

- Please specify what you included under the term "postoperative complications" in your methods.

- Please specify the management of R1 cases in your institution. Please specify the chemotherapy protocols used in your institute for chest wall tumors in the methods section.

- Line 63: please add this reference in the reference section (rather than text).

- Line 73: Please revisit and confirm whether you mean Trucut biopsy vs fine needle aspiration biopsy. If "fine needle aspiration biopsy" was your institutional approach, please discuss advantages and disadvantages in the discussion section.

- Line 91: please report interquartile range for any reported median rather than range. Please add the word "respectively" to this sentence "... standard deviation (SD) or median (range) for normally and non-normally distributed quantitative variables."

- Line 123: if possible: please add more intra-operative picture especially myocutaneous flaps reconstruction in a single figure that include many panels.

- Line 128: Please specify why you used 65mm as a cutoff. Was it chosen arbitrary/based on a prior publication/mean or median of tumor size, etc? and mention that in your methods section. Same comment for choosing 2 ribs in your models.

- Line 137: Please recheck p value as it should be significant given that 95%CI doesn't include 1. P should be <0.05.

- Line 141: please be consistent in referring to tables/figures eg "(table n. 4, figure n. 2)." should be "(Table 4, Figure 2)."

- Please add "Strength and limitations" section at the end of discussion and mention that this work represents small sample size in the limitation.

- Table 1 and 2: please revisit histology rows (numbers) as you mentioned that sarcoma represents in 51 sarcoma in table 1 vs 49 sarcomas in table 2. Please add Log-rank test P value (not Chi square test) as you are reporting time to events (death) outcome (long term mortality). Optional request: you may add hazard ratio (95%CI) and p value as well using Cox regression. Please edit "Hysyollogy" to "Histology". Please edit "Metil-methacrylate" to "Methyl-methacrylate"

- Please report goodness of fit for your multivariate analysis for OS and RFS.

- Please specify median follow up time in your cohort.

Please try to do Kaplan Meier curves for A) Ewing's sarcoma vs others and for B) sternal reconstruction vs others. You may add demographic criteria of those groups in the supplements (optional request).

Author Response

Lo Iacono et al reported their work named "The role of surgery in primary chest wall tumors: over 20 years’ experience in resection and reconstruction " and concluded "Treatment of primary chest wall tumors remains a very challenging topic. Different histological types do not help in defining a univocal approach. Complete resection with healthy margins remains a definitive cornerstone in the treatment of these cancers as part of a more comprehensive treatment.". I have the following comments:

- Please add grouped bar plot with annual cases and Ewing's sarcoma cases over time adjacent to each other in the same figure.

Answer: thank you for your comment. The requested graph does not add to the manuscript. We could eventually add a scatter plot.

Change: none.

- Please specify what you included under the term "postoperative complications" in your methods.

Answer: thank you for your comment. We better specify the term.

Change: in red colour.

- Please specify the management of R1 cases in your institution. Please specify the chemotherapy protocols used in your institute for chest wall tumors in the methods section.

Answer: thank you for your comment. We specify the management of R1 cases and the chemotherapy protocols used.

Change: in red colour.

- Line 63: please add this reference in the reference section (rather than text).

Answer: thank you for your comment. It was a typo; we added as reference.

Change: in red colour.

- Line 73: Please revisit and confirm whether you mean Trucut biopsy vs fine needle aspiration biopsy. If "fine needle aspiration biopsy" was your institutional approach, please discuss advantages and disadvantages in the discussion section.

Answer: thank you for your comment. We had better discuss this point.

Change: in red colour.

- Line 91: please report interquartile range for any reported median rather than range. Please add the word "respectively" to this sentence "... standard deviation (SD) or median (range) for normally and non-normally distributed quantitative variables."

Answer: thank you for your comment. We have included your suggestions.

Change: in red colour.

- Line 123: if possible: please add more intra-operative picture especially myocutaneous flaps reconstruction in a single figure that include many panels.

Answer: thank you for your comment. We have included in the manuscript all possible pictures.

Change: none.

- Line 128: Please specify why you used 65mm as a cutoff. Was it chosen arbitrary/based on a prior publication/mean or median of tumor size, etc? and mention that in your methods section. Same comment for choosing 2 ribs in your models.

Answer: thank you for your comment. The cut-off of 65 mm was chosen based on mean of tumor size; the cut-off of 2 ribs was similarly chosen based on the mean value of resected ribs. We added in the methods section.

Change: in red color.

- Line 137: Please recheck p value as it should be significant given that 95%CI doesn't include 1. P should be <0.05.

Answer: thank you for your comment. There was a typo on the table that we had amended.

Change: in red color.

- Line 141: please be consistent in referring to tables/figures eg "(table n. 4, figure n. 2)." should be "(Table 4, Figure 2)."

Answer: thank you for your comment. There was a typo on the table that we had amended.

Change: in red color.

- Please add "Strength and limitations" section at the end of discussion and mention that this work represents small sample size in the limitation.

Answer: thank you for your comment. We added the Strength and limitations section.

Change: in red color.

- Table 1 and 2: please revisit histology rows (numbers) as you mentioned that sarcoma represents in 51 sarcoma in table 1 vs 49 sarcomas in table 2. Please add Log-rank test P value (not Chi square test) as you are reporting time to events (death) outcome (long term mortality). Optional request: you may add hazard ratio (95%CI) and p value as well using Cox regression. Please edit "Hysyollogy" to "Histology". Please edit "Metil-methacrylate" to "Methyl-methacrylate"

Answer: thank you for your comment. There were typos on the tables that we had amended.

Change: in red color.

- Please report goodness of fit for your multivariate analysis for OS and RFS.

Answer: thank you for your comment. We added GOF.

Change: in red color.

- Please specify median follow up time in your cohort.

Answer: thank you for your comment. We add the media follow-up time.

Change: in red color.

Please try to do Kaplan Meier curves for A) Ewing's sarcoma vs others and for B) sternal reconstruction vs others. You may add demographic criteria of those groups in the supplements (optional request).

Answer: thank you for your comment. We add the requested KM curves.

Change: in red color.

Reviewer 2 Report (Previous Reviewer 1)

Dear authors,

thank you very much for your responses an changes in the manuscript.

Author Response

Thank you.

Round 2

Reviewer 1 Report (Previous Reviewer 3)

Authors thankfully addressed my prior comments apart from 3 points that are still not answered and need their attention as I can't see proper reply in the manuscript regarding them:

- Please add grouped bar plot with annual cases and Ewing's sarcoma cases over time adjacent to each other in the same figure.

- Line 73: Please revisit and confirm whether you mean “Trucut biopsy/Core biopsy” rather than "cytology" obtained via “fine needle aspiration biopsy”. If "fine needle aspiration biopsy" was your institutional approach, please discuss advantages and disadvantages in the discussion section.

- Please report goodness of fit for your multivariate analysis for OS and RFS.

Author Response

Authors thankfully addressed my prior comments apart from 3 points that are still not answered and need their attention as I can't see proper reply in the manuscript regarding them:

- Please add grouped bar plot with annual cases and Ewing's sarcoma cases over time adjacent to each other in the same figure.

Answer: please find the requested plot.

Changes: in red color.

- Line 73: Please revisit and confirm whether you mean “Trucut biopsy/Core biopsy” rather than "cytology" obtained via “fine needle aspiration biopsy”. If "fine needle aspiration biopsy" was your institutional approach, please discuss advantages and disadvantages in the discussion section.

Answer: we revisit the paragraph.

Changes: in red color.

- Please report goodness of fit for your multivariate analysis for OS and RFS.

Answer: the goodness of fit was added for the multivariable analysis.

Changes: in red color

This manuscript is a resubmission of an earlier submission. The following is a list of the peer review reports and author responses from that submission.

Round 1

Reviewer 1 Report

Dear authors,

I was allowed to review your manuscript. The topic is interesting and important. The retrospective analysis covers a long period of time with relatively few patients, although this is certainly due to the rarity of space-occupying lesions in the thoracic wall. I have the following questions and comments.

1. please describe in more detail in the results which reconstruction techniques you used. Were additional flap plastics necessary to cover larger defects? Cooperation with plastic surgery?

2. the Kaplan-Meier curves are copied from SPSS. I recommend to improve the curves in layout.

3. table 3: ICU-stay, please correct.

4. table 4, legend: analysis, please correct.

5. line 190: lymphadenectomy, please correct.

6. a figure/flow-chart of your therapy approach would certainly be interesting for the reader. When do you do which biopsy, what staging, when neoadjuvant therapy, when direct surgery, when adjuvant therapy....

7. it should also be made clear that these rare tumors are mostly sarcomas and should be treated in specialized centers and interdisciplinary.

Author Response

1. please describe in more detail in the results which reconstruction techniques you used. Were additional flap plastics necessary to cover larger defects? Cooperation with plastic surgery?

We explained this aspect into the text.

“Should be paid attention to the impaction of the scapula for the resection in the posterol-ateral resection. In these cases, a synthetic mesh must necessarily be placed. In 20 patients it was necessary to proceed with a reconstruction with a rotational muscle flap in order to cover the prosthesis. The most used muscle was the latissimus dorsi in 15 cases, while in the remaining 5 cases we created a pectoralis major muscle flap. All reconstructions were performed by our plastic surgeons”.

2. the Kaplan-Meier curves are copied from SPSS. I recommend to improve the curves in layout.

We modified and improved Kaplan-Meier curves and we added patients at risk (figure n. 1 and 2)

3. table 3: ICU-stay, please correct.

Done.

4. table 4, legend: analysis, please correct.
Done.

5. line 190: lymphadenectomy, please correct.
Done.

6. a figure/flow-chart of your therapy approach would certainly be interesting for the reader. When do you do which biopsy, what staging, when neoadjuvant therapy, when direct surgery, when adjuvant therapy....
The suggestion is very interesting, however we believe that it is not feasible to be able to frame everything strictly in a flowchart, because the heterogeneity of the pathology and the numerous histologies do not help us to give unambiguous answers along the way, leaving many of the choices of the path to the multidisciplinary meeting . The only decisive parameter, as specified in the text, can be considered the size, and therefore small lesions can be removed directly without preoperative diagnosis.

7. it should also be made clear that these rare tumors are mostly sarcomas and should be treated in specialized centers and interdisciplinary.

Added at the end of the manuscript (conclusions)

Given the particularity of the surgery and the rarity of the presentation, it is important that patients are patients must be referred and treated in high-volume centers with more expe-rience in the field and with the presence of a multidisciplinary team, which can provide the patient with all the necessary tools for care in all its aspects.”

Thank you.

Reviewer 2 Report

Nice article, great work on patients. 

Changes in English language and style need to be addressed in this article. 

Line 23 and 119 --> should be corrected to " remaining 5 " 

line 39- should be corrected to " dominated by sarcomas" 

line 41 -should be corrected to " patients typically refer to the appearance of a mass" 

line 42- should be corrected to"non specificity of initial symptoms do not help"

line 46 and 70- should be corrected to " neoadjuvant" 

line 58- should be corrected to " primary chest wall " instead of primitive

line 108- should be corrected to " consisted of" 

line 109- 1 patient ( not patients) 

table n4- " analysis" spelling needs to be corrected

Not sure what is the meaning of " Metil-met" in the table

Spelling of sarcoma needs to be corrected . Its mentioned as sarcome under histology in the tables. 

line 167- word " fever" repeated twice

line 186- needs to be corrected 

line 190- believe authors wanted to say " lymphadenectomy" 

line 216- should be corrected to " always need" instead of " need always"

Author Response

Nice article, great work on patients. 

Changes in English language and style need to be addressed in this article. 

We perform an extensive English revision for the article.

Line 23 and 119 --> should be corrected to " remaining 5 " 
Done

line 39- should be corrected to " dominated by sarcomas" 
Done

line 41 -should be corrected to " patients typically refer to the appearance of a mass" 
Done

line 42- should be corrected to"non specificity of initial symptoms do not help" 
Done

line 46 and 70- should be corrected to " neoadjuvant" 
Done

line 58- should be corrected to " primary chest wall " instead of primitive

Done

line 108- should be corrected to " consisted of" 
Done

line 109- 1 patient ( not patients) 
Done

table n4- " analysis" spelling needs to be corrected

Done

Not sure what is the meaning of " Metil-met" in the table
It refers to "methyl methacrylate". Corrected and added as footnote.

Spelling of sarcoma needs to be corrected . Its mentioned as sarcome under histology in the tables. 

Done

line 167- word " fever" repeated twice
Done

line 186- needs to be corrected 
Done

line 190- believe authors wanted to say " lymphadenectomy" 
Done

line 216- should be corrected to " always need" instead of " need always"
Done

Reviewer 3 Report

Lo Iacono et a l presented his work entitled "The role of surgery in primary chest wall tumors: over 20 years’ 2 experience in resection and reconstruction" and concluded the following "Treatment of primary chest wall tumors remains a very challenging topic. Different histological types do not help in defining a univocal approach. Complete resection with healthy margins remains a definitive cornerstone in the treatment of these cancers, both as an exclusive treatment and as part of a more comprehensive treatment.". I have the following comment:

- Language revision is essential 

- In Simple Summary: please delete this " as the only treatment" in this sentence "However, we can say that oncological radical surgery is an essential element as the only treatment and even in multidisciplinary paths.".

-  Please delete this word "both as an exclusive treatment and " from this sentence "Complete resection with healthy margins remains a definitive cornerstone in the treatment of these cancers, both as an exclusive treatment and as part of a more comprehensive treatment.".

- Please add referenc eto this sentence "It’s believed that Osias Aimar was the first to perform, in 1778,"

- A prior paper reported that lower blood loss was a predictor of better survival (https://pubmed.ncbi.nlm.nih.gov/29312725/). Please comment on that and add it to limitation if you are unable to investigate this in the current work.

- Table 1 and 2: Please add percent beside each number out of the total number in each "column". Please change "," in P value column to ".". Please change "sarcome" to "sarcoma". Please change "Multi-met" to whatever looks appropriate in the reconstructive prothesis and enumerate the used reconstruction approaches in the table footnote or in the results section. Please enumerate the 30 days morbidities reported in Table 2 as a supplementary table. Please add a row for 30 days mortality to your Table 2. Please add total number of patients included in each column in the heading eg dead (n=16)

- For table 3: Please report hazard ratio and its 95% confidence interval, and P values ONLY. No need to report "beta". Generally, HR= exponential beta. Please specify selection criteria for factors selection into multivariate analysis (MVA). You have only 53 patients with 16 mortality, so you can include only 2 factors in MVA for mortality and you have 23 patients who developed relapse, so you can include 2-3 variables in MVA for relapse.

- Reported cumulative OS below table 4 could be deleted or moved to supplementary material with addition of more variables as relapse, gender, grade...etc. This is optional request).

- Please add patients at risk to Kaplan Meier curves in Figure 1. Please add the current figure 2 to supplements.

- Figure 5C: please identifies the added landmarks in the legend eg Specimen with landmarks where IVR is right 4th rib.

Author Response

- Language revision is essential

We perform an extensive English revision for the article.

- In Simple Summary: please delete this " as the only treatment" in this sentence "However, we can say that oncological radical surgery is an essential element as the only treatment and even in multidisciplinary paths.".
Done

-  Please delete this word "both as an exclusive treatment and " from this sentence "Complete resection with healthy margins remains a definitive cornerstone in the treatment of these cancers, both as an exclusive treatment and as part of a more comprehensive treatment.".
Done

- Please add referenc eto this sentence "It’s believed that Osias Aimar was the first to perform, in 1778,"
Done.

- A prior paper reported that lower blood loss was a predictor of better survival (https://pubmed.ncbi.nlm.nih.gov/29312725/). Please comment on that and add it to limitation if you are unable to investigate this in the current work.

We added this aspect in the discussion

“A series published in 2017 showed as predictors of better DFS a lower blood loss during surgery. (16) Although we have not collected this data to evaluate it, as we have never had the inkling of large losses during surgery, given the works that show a likely significance, it could be a data to consider for future work”.

- Table 1 and 2: Please add percent beside each number out of the total number in each "column". Please change "," in P value column to ".". Please change "sarcome" to "sarcoma". Please change "Multi-met" to whatever looks appropriate in the reconstructive prothesis and enumerate the used reconstruction approaches in the table footnote or in the results section. Please enumerate the 30 days morbidities reported in Table 2 as a supplementary table. Please add a row for 30 days mortality to your Table 2. Please add total number of patients included in each column in the heading eg dead (n=16)

Done. We changed “multi-met” into “m. methacr” and we added in the footnotes the name “metyl methacrylate”. We do not report 30-days mortalities because we did not reported any 30-day death.

- For table 3: Please report hazard ratio and its 95% confidence interval, and P values ONLY. No need to report "beta". Generally, HR= exponential beta. Please specify selection criteria for factors selection into multivariate analysis (MVA). You have only 53 patients with 16 mortality, so you can include only 2 factors in MVA for mortality and you have 23 patients who developed relapse, so you can include 2-3 variables in MVA for relapse.

Done. Selection criteria for factors selection into MVA was the significance of these variables at univariate analysis (both for OS and DFS). These significant variables at univariate analysis, were entered in a Cox multivariate analysis to assess their independent character. Significance was accept-ed at a level of less than 0.05

- Reported cumulative OS below table 4 could be deleted or moved to supplementary material with addition of more variables as relapse, gender, grade...etc. This is optional request).
Thank you for your advice.

- Please add patients at risk to Kaplan Meier curves in Figure 1. Please add the current figure 2 to supplements.

We modified and improved Kaplan-Meier curves and we added patients at risk (figure n. 1 and 2)

- Figure 5C: please identifies the added landmarks in the legend eg Specimen with landmarks where IVR is right 4th rib.
Done